# Is Adolescents’ Free Sugar Intake Associated with the Free Sugar Intake of Their Parents?

**DOI:** 10.3390/nu14224741

**Published:** 2022-11-10

**Authors:** Rou Zhang, Qiping Yang, Qiong Tang, Yue Xi, Qian Lin, Lina Yang

**Affiliations:** Xiangya School of Public Health, Central South University, Changsha 410128, China

**Keywords:** free sugar, adolescents, parents, influencing factors

## Abstract

High free sugar intake can lead to increased dental caries, obesity, and other health risks among adolescents. Studies have shown that family factors, especially parents, are one of the primary factors influencing adolescents’ sugar intake. This study aims to investigate the influence of adolescent parents’ free sugar intake, knowledge, attitude, and practice (KAP) on adolescents’ free sugar intake. A total of 1090 pairs of adolescents and their parents from 10 secondary schools in Changsha were enrolled in a cross-sectional study. Free sugar intakes of parents and adolescents were measured using the food frequency questionnaire (FFQ). The current status of parents’ knowledge, attitude, and practice in consuming free sugar was investigated using online and offline questionnaires. Parental free sugar intake was 11.55 (5.08, 21.95) g/d, and that of adolescents was 41.13 (19.06, 80.58) g/d. Parental free sugar intake, free sugar knowledge level, intake behavior, and guidance behavior were associated with adolescent free sugar intake. A superior level of parental free sugar knowledge (adjusted OR = 0.726, 95% CI: 0.557~0.946) was a protective factor for adolescent free sugar intake. Moderate and high levels of parental free sugar intake (adjusted OR = 1.706, 95% CI: 1.212~2.401; adjusted OR = 2.372, 95% CI: 1.492~3.773, respectively) were risk factors for free sugar intake in adolescents. Given the importance of parental influence on the adolescent free sugar intake, further limiting parental intake and increasing awareness of free sugars could play an active role in future interventions for adolescents’ free sugar intake.

## 1. Introduction

A high intake of free sugar is considered to be one of the main contributors to the increased risk of obesity in children and adolescents [1,2]. One study suggests that dietary sugars are associated with increased obesity and a subsequent increased risk of diet-related diseases when consumed as an excessive source of calories [3]. Several studies have shown that children and adolescents with a free sugar intake exceeding 10% of their total energy intake have a higher risk of dental caries than those with an intake below 10% [4,5,6]. Previous studies have demonstrated that excessive free sugar intake (above 10% of the total energy intake) increases the risk of dental caries, pharyngeal reflux, and excessive daytime sleepiness [7,8,9], as well as increasing the risk of cardiovascular disease [10,11] and other related health problems. According to the definition of free sugars in the Guidelines for Sugar Intake in Adults and Children, free sugars are all sugars other than naturally occurring monosaccharides and disaccharides, i.e., all sugars added to foods by manufacturers, chefs, or consumers and naturally occurring in honey, syrups, fruit juices, and fruit juice concentrates [12,13,14].

The World Health Organization highlighted the recommended intake of free sugars and proposed limiting the intake of free sugars to less than 10% of total energy intake in adults and children, and further limiting it to less than 5% [12,15]. Free sugars include added sugars, and the largest source of added sugar intake among adolescents is sugar-sweetened beverages (SSBs) [16]. It has been confirmed that the high consumption of SSBs by adolescents is the most significant cause of high free sugar intakes, and often leads to adolescents becoming high consumers of free sugars. The absolute intake of free sugars in Canadian adolescents was 80 g/d, which is 1.5 times higher than that of adults [17]. Free sugar intake among European adolescents was higher than 87 g/d [18]. Additionally, a survey on the free sugar content of commercially available beverages in China in 2015 showed that 78.4% of beverages contained ≥5 g/100 g of free sugar, which was at a high level [19]. Subsequently, an assessment of free sugar intake in the general urban population in 2018 showed that the adolescent population had the largest contribution to free sugar intake through SSBs at 40.73 g/d [20].

Substantial high consumption of free sugars and added sugars in adolescents has become an increasingly prominent global public health concern due to the negative health effects of high-sugar diets [21,22,23,24,25,26]. Several regions have started to pay gradual attention to the factors associated with sugar intake affecting adolescents. In recent years, several studies on factors associated with adolescent sugar intake have shown that family factors, environmental factors, and adolescents’ own factors are associated with increased adolescent sugar intake. Studies have shown that children and adolescents have an innate preference for sweet flavors [27], but there are other influences that have been affecting their preference for sweets in recent years. Studies on dietary factors related to obesity in adolescents have found that the socio-cultural environment, especially parental behavior and family influence, has a greater impact on adolescent dietary behavior and intake; that is, parental dietary intake is consistently correlated with adolescent dietary intake [28,29,30].

During adolescence, lifestyles and behaviors become more autonomous, and changes in social and environmental factors have a great impact on their autonomy. Numerous studies have shown that family, especially parental behavior, is one of the main factors influencing adolescents’ sugar intake [31]. Acquiring knowledge, generating beliefs, and forming behaviors can be summarized as knowledge, attitude, and practice (KAP), which are important factors in explaining health-related behaviors and also influence each other [32]. Recently, a cross-sectional study was used to investigate maternal KAP of free sugar and its association with free sugar intake in children aged 6–12 years in Saudi Arabia [32]. In China, there are no studies on the relationship between adolescent parental free sugar intake, KAP, and adolescent free sugar intake at this time. Therefore, the aim of this study was to investigate the association between parental free sugar intake, free sugar KAP, and adolescents’ free sugar intake.

## 2. Materials and Methods

### 2.1. Ethical Approval

The investigation was advanced with the approval of the Education Bureau and was approved by the Ethical Review Committee of Xiangya School of Public Health, Central South University (XYGW-2019-025).

### 2.2. Study Design and Sampling

A multi-stage cluster random sampling method was used in this study. After we obtained approval from the Education Bureau, 10 secondary schools in 5 administrative districts within Changsha were randomly selected, with 2 secondary schools randomly selected from each district; in each school, two classes were randomly selected from each of the first and second grade, and all students in the class and their parents were included in the survey. Before the formal investigation, parental informed consent was obtained through a paper informed consent form. The two sections of the study included parent and student questionnaires, which were completed by parents and students, respectively.

Based on the inclusion and exclusion criteria, students and their parents were included in the sample if they qualified as follows: (1) the number of students in the first and second grade of school was at least 500 and (2) students’ parents gave their consent to participate in the survey. The exclusion criterion was the inability to complete the questionnaire because of an inability to read or write. Eligible students and their parents were included in the study sample based on inclusion and exclusion criteria. A total of 1628 students participated in the study, 1517 of whom were surveyed. A total of 1136 parents were surveyed. Parent and student data were matched, and non-completed questionnaires were excluded, and the number of student and corresponding parent sample cases included in the final analysis was 1090 pairs.

### 2.3. Questionnaire Survey

Before the formal survey, informed consent and parental questionnaires were distributed to students in the form of letters, which were taken home by students for their parents to fill in informed consent and choose electronic or paper versions of the questionnaires to complete. Students were instructed by uniformly trained personnel to fill out the online questionnaire in the school computer room. The following information were collected:Demographic information: parent–student kinship, gender, age, Body Mass Index (BMI), and ethnicity; parent education, average monthly household income, and student pocket money.Free sugar intake of parents and students: free sugar intake of parents or students in the past month was collected by semi-quantitative FFQ. The semi-quantitative FFQ in our study was developed based on two previous studies conducted by the Chinese Center for Disease Control and Prevention (CDC) [33,34]. The options of intake frequency were set as never eat/drink, monthly frequency, weekly frequency, and daily frequency. To assess the total daily free sugar intake of individuals, the frequency of intake was all converted to daily frequency of intake, converting 1 time per week to 0.14 times/day, 2–3 times per week to 0.36 times/day, etc. In this study’s questionnaire, free sugar types included SSBs (carbonated beverages, tea drinks, fruit and vegetable juice drinks, plant protein drinks, functional drinks, milk tea, and flavored sour milk) and sugary foods (sugary snacks, candy, dried fruit, and honey).Parental free sugar KAP [35]: parents’ knowledge of free sugar, parents’ attitude toward controlling adolescents’ free sugar intake, parents’ behavior regarding the intake of sugary foods, purchase and storage behavior of beverages and desserts, and parents’ guidance behavior toward adolescents were considered.Classification of KAP: we divided the respondents into two categories based on the 75th percentile of their knowledge, attitude, and practice scores. Values less than P75 were defined as low levels of free sugar knowledge, negative attitudes, and poor behavior. On the contrary, others were defined as high levels of free sugar knowledge, positive attitudes, and better behaviors.

### 2.4. Free Sugar Intake Assessment

The following formula was used to calculate the free sugar intake:$$Z=A\left(f_{1}\right)\times A\left(m_{1}\right)\times c_{1}+B\left(f_{2}\right)\times B\left(m_{2}\right)\times c_{2}+\cdots +X\left(f_{n}\right)\times X\left(m_{n}\right)\times c_{n}$$

Total individual free sugar intake per day (Z, g/day); frequency of food intake (f, times/day); food consumption (m, g or mL); food sugar content coefficient referencing data from the Chinese Center for Disease Control and Prevention [34] (c, g/100 g or g/100 mL); food types (A, B, … X); natural number (n). Based on the above formula, the data of daily free sugar intake for parents and adolescents could be obtained. According to the recommendations of the latest version of the Dietary Guidelines for Chinese Residents (2022) [36], the free sugar intakes of parents and adolescents were divided into low-, medium-, and high-free-sugar-intake groups, with 25 g and 50 g as the threshold values.

### 2.5. Statistical Analysis

Data were double-recorded and tested for consistency using EpiData 3.1 software (The EpiData Association, Odense, Denmark). SPSS26.0 statistical software (IBM Corp., Armonk, NY, USA) was used for data analysis, and GraphPad Prism8.0 software (GraphPad Software, Inc., San Diego, CA, USA) was used for plotting. As the data were distributed in ranks, statistical methods included Mann–Whitney U test and Kruskal–Wallis H test for nonparametric tests, Spearman’s rank correlation analysis, and ordinal logistic regression analysis. In this study, measurement data were described using median and quartiles (Median (P25, P75)), and count data were described as percentages (%). *p*-values < 0.05 were considered to indicate statistical significance.

## 3. Results

### 3.1. Characteristics of the Study Sample

A total of 1090 adolescent–parent pairs participated in this survey. Among them, there were 569 adolescent boys (52.2%) and 521 adolescent girls (47.8%) (Table 1). The average age of the surveyed adolescents was 13.54 ± 0.64 years. The parents of the adolescents who participated in the survey comprised 326 (29.9%) males and 764 (70.1%) females, with an average age of 41.85 ± 5.28 years.

### 3.2. Parental Free Sugar Intake Characteristics

According to the results of the study, the parental free sugar intake was 11.55 (5.08, 21.95) g/d. The Dietary Guidelines for Chinese Residents (2022) recommends that the appropriate daily intake of sugar for adults be limited to less than 25 g and not more than 50 g at most. The results of our study showed that 79.0% of parents consumed less than 25 g/d of free sugar, and 7.5% of parents consumed more than the maximum intake level of 50 g/d (Table 2). The results of the univariate analysis show that the younger parents consumed more free sugar (*p* < 0.05). There were also statistical differences in the level of free sugar intake in terms of parents’ attitude and practice (including intake behavior and guidance behavior) toward free sugar (*p* < 0.05).

### 3.3. Association between Parental Free Sugar Intake and Adolescents’ Free Sugar Intake

According to the results of our study, adolescents’ free sugar intake was 41.13 (19.06, 80.58) g/d, with 32.2% of adolescents consuming less than 25 g/d and 41.6% consuming more than the recommended maximum intake level of 50 g/d (Table 3). The correlation analysis showed that parental free sugar intake levels showed a positive correlation with adolescent free sugar intake levels (r_s_ = 0.159, *p* < 0.001). The correlation analysis of free sugar intake between parental and adolescents’ free sugar intake showed a positive correlation (r_s_ = 0.143, *p* < 0.001), as shown in Figure 1.

### 3.4. Correlation between Parental Free Sugar KAP Levels and Adolescents’ Free Sugar Intake

The correlation analysis of free sugar intake levels showed that the level of parental free sugar knowledge and total intake behavior were negatively correlated with adolescent free sugar intake (correlation coefficients r_s_ were −0.087, −0.100, *p* < 0.01, respectively). Among the total behaviors of parental free sugar intake, parental free sugar intake behavior and parental guidance behavior on adolescent free sugar intake showed negative correlations with adolescent free sugar intake (correlation coefficients r_s_ were −0.062, −0.085, *p* < 0.05, respectively). Parents’ attitudes toward controlling adolescents’ free sugar intake and parents’ purchase and storage behavior were not correlated with adolescents’ free sugar intake (*p* > 0.05) (see Table 4).

### 3.5. Adolescents’ Free Sugar Intake in Terms of Parental Free Sugar Intake in Ordinal Logistic Regression

The adolescents’ free sugar intake grouping (low, medium, and high) was used as the dependent variable, and the parents’ gender, role, age, education level, monthly household income, and parental free sugar KAP-related variables and free sugar intake level were included as independent variables. Our group studied the correlation between adolescent free sugar intake and dental caries, pharyngeal reflux, and excessive daytime sleepiness [7,8,9]. The studies found that adolescents’ age, grade level, and reading style were related to their free sugar intake. So, we considered including adolescents’ age, grade level, and reading style in the variables as adjustment factors, and conducted an ordinal logistic regression analysis to assess the relationship between parental-free-sugar-intake-related factors and adolescent free sugar intake. The results of the parallel line test were $\chi^{2}$ = 24, *p* = 0.355, indicating that the model met the proportional dominance hypothesis. In model 1, compared with low levels of free sugar intake, moderate and high levels of parental free sugar intake (OR = 1.760, 95% CI: 1.256~2.465; OR = 2.369, 95% CI: 1.500~3.744, respectively) were more likely to increase adolescent free sugar intake and were risk factors for high free sugar intake in adolescents (Table 5). Compared to parents with poorer knowledge of free sugars, those with better knowledge (OR = 0.720, 95% CI: 0.555~0.934) were more likely to reduce their adolescents’ intake of free sugars. These associations persisted after adjusting for multiple variables. The results show that after correcting for factors, moderate and high levels of parental free sugar intake (adjusted OR = 1.706, 95% CI: 1.212~2.401; adjusted OR = 2.372, 95% CI: 1.492~3.773, respectively) were risk factors for adolescent free sugar intake relative to low levels of free sugar intake. Additionally, better parental knowledge of free sugars (adjusted OR = 0.726, 95% CI: 0.557~0.946) was a protective factor for free sugar intake in adolescents.

## 4. Discussion

The study found that the free sugar intake of adolescent parents was 11.55 (5.08, 21.95) g/d, with 92.5% of the parents consuming less free sugar than recommended by WHO (<10 E%). The free sugar intake of the parents was lower than free sugar intake in other countries, such as Australia, Greece, etc. [37]. We found that the free sugar intake was higher among people under 40 years old in different age groups, which is similar to the results of the assessment of sugary beverage consumption and free sugar intake of urban residents in China in 2018 [20]. The free sugar intake of adolescents in this survey was 41.13 (19.06, 80.58) g/d, which far exceeds the measured 22.5 g/d (mean) of added sugar intake among Chinese adolescents aged 12–17 in China in 2012 and 40.73 g/d (P95 value) of free sugar intake in SSBs in 2018 [20,33]. In our study, 41.6% of adolescents had free sugar intakes of more than 50 g/d. The status of adolescents’ free sugar intake is not optimistic. These studies indicate that the sugar intake of adolescents in China shows a rising trend, and the current situation of the high free sugar and added sugar intakes of adolescents deserves further attention. Therefore, prevention and control of adolescent sugar intake is of great relevance. We found in this study that higher parental free sugar intake had a greater impact on adolescent free sugar intake. It is a feasible approach to developing an intervention from the perspective of adolescent parents.

Our study found that three aspects of parental free sugar KAP correlates—the level of knowledge of free sugar, intake behavior, and guidance behavior—were significantly associated with adolescents’ free sugar intake. Norwegian parents’ knowledge of SSBs was a significant influencing factor for adolescents’ SSBs knowledge in the region [38]. In our results, parental knowledge of free sugars was not correlated with their own free sugar intake, but it influenced adolescents’ free sugar intake, and ordinal logistic regression analysis also showed that high parental free sugar knowledge was a protective factor for adolescent free sugar intake. In a cross-sectional study in Saudi Arabia, mothers were found to be making efforts to reduce their children’s free sugar intake despite their limited level of knowledge about free sugars. That is, mothers’ attitude and practice regarding free sugars affect free sugar intake in children and adolescents aged 6–12 years [32]. Previous studies have also shown higher free sugar knowledge level in female parents [35]. This shows that future interventions in improving parents’ nutritional knowledge (including free sugars) have a high potential to influence adolescents’ free sugar intake.

The dietary behavior of parents, especially caregivers, is a major factor influencing children’s dietary behavior [39]. In this study, we found that parents with better free sugar intake behavior had a lower free sugar intake, and their children had lower free sugar intakes. Parental guidance behavior related to free sugars further indicated their positive influence on adolescents’ free sugar intake. Consistent with previous studies, parents had a major impact on adolescents’ overall nutritional intake by reinforcing healthy eating concepts, mimicking eating behaviors, and providing food [40,41]. In contrast, we observed that there was no association between purchase storage behavior and adolescent free sugar intake, which is consistent with a study on the non-association between the availability of dairy beverages among African American and Hispanic parents and children’s dairy beverage intake [42]. A study on the effects of a low-free-sugar diet on adolescent nonalcoholic fatty liver disease (NAFLD) showed that when a dietary intervention was administered to the entire family (including an intervention that restricted the family from purchasing any food), preliminary results show that a low-free-sugar diet improved NAFLD risk [43]. These results suggest that parental behaviors in guiding the diet of the family have an impact on their children’s diet.

The sugary food intake of adolescent parents has more influence on children’s intake than other characteristics [44]. Studies have shown that parental SSB intake can reflect adolescents’ SSB consumption to some extent [45]. In our study, there was a positive association between adolescent parental free sugar intake and children’s free sugar intake. Regression results show that a high level of parental free sugar intake was a risk factor for increased free sugar intake in children. Other studies have also observed that parental food (e.g., beverage) intake is associated with food intake in children or adolescents [46,47,48]. These results emphasize the importance of parents, in particular, as intervention targets to reduce free sugar intake in adolescents [21,49].

Given the burden of high-sugar diets on adolescents’ health, it is essential to implement as many relevant interventions as possible, including within the family, school, and society. To further limit the intake of SSBs among adolescents, “traffic light” labeling in the United Kingdom and front-of-pack labeling for foods with a high content of sugar or other ingredients in Mexico have been adopted, which enable adolescents to effectively identify SSBs and achieve the effect of controlling consumption [50]. Our country (China) focused on the “three reductions and three health” in the National Action Plan for a Healthy Lifestyle (2017–2025) [51] to reduce the intake of added sugar. Parents, as an important factor in family environment factors, have a significant influence on the diet and nutrition of children and adolescents [52]. Therefore, it is necessary to carry out targeted scientific guidance, pay attention to the publicity of free-sugar-related knowledge, and adolescents’ parents’ guidance of intake behavior, so as to provide a strong guarantee for reducing the total amount of free sugar consumed by adolescents under family environment-related factors. At the same time, we should enhance nutritional health education for adolescents in school to raise their awareness of free sugar intake level, reduce their high intake behaviors, and gradually help them to develop reasonable dietary habits and healthy lifestyles. At present, there is no study on parental factors related to free sugar intake among Chinese adolescents. This study was the first to investigate the effects of parental free sugar intake and KAP on free sugar intake among adolescents in Changsha, China, using parents of adolescents as the main influencing family factor. However, the limitation of our study is that it solely investigated the Changsha area, as the dietary habits and food culture are different in each region of China. Currently, studies on free sugar intake in adolescents are scarce; so, in the future, it is recommended that studies be conducted in other areas.

## 5. Conclusions

Our study showed that the level of free sugar intake, free-sugar-related knowledge, and guidance behavior of adolescents’ parents in Changsha were associated with adolescent free sugar intake. Thus, parental knowledge and behavior of free sugars have an important role in free sugar intake among adolescents, and developing interventions for parents may help to effectively reduce adolescents’ intake of free sugar. Therefore, future interventions on adolescents’ free sugar intake in China could be implemented in terms of parental free sugar intake and to help them understand the multiple factors influencing adolescents’ free sugar intake to effectively reduce their intake.

## Figures and Tables

**Figure 1 nutrients-14-04741-f001:**
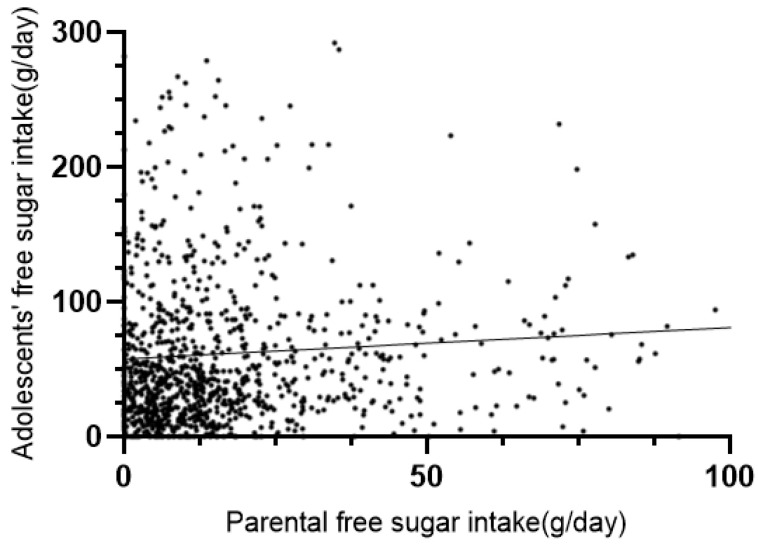
Association between parents and adolescents’ free sugar intake.

**Table 1 nutrients-14-04741-t001:** Characteristics of parents and adolescents (*n* = 1090).

Characteristics	Parents		Adolescents	
	Sample Size (*n*)	Percentage (%)	Sample Size (*n*)	Percentage (%)
Gender				
Male	326	29.9	569	52.2
Female	764	70.1	521	47.8
Parental role				
Father	315	28.9	-	-
Mother	757	69.4	-	-
Grandparents	18	1.7	-	-
Age ^#^	1090	41.85 ± 5.28	1090	13.54 ± 0.64
Ethnicity				
Hans	1051	96.4	1051	96.4
Minorities	39	3.6	39	3.6
Parental education				
≤Primary school	25	2.3	-	-
Middle school	261	23.9	-	-
(Vocational) high school	465	42.7	-	-
≥College	339	31.1	-	-
Family monthly income				
CNY ≤ 1000	20	1.8	-	-
CNY 1001–3000	193	17.7	-	-
CNY 3001–5000	299	27.4	-	-
CNY 5001–7000	223	20.5	-	-
CNY ≥7001	355	32.6	-	-
Student pocket money				
Low	-	-	1026	94.1
Medium	-	-	27	2.5
High	-	-	32	2.9
BMI				
Wasting	62	5.7	42	3.9
Normal	735	67.4	755	69.3
Overweight	249	22.8	130	11.9
Obese	44	4.0	163	15.0

^#^ Age data are expressed as mean ± SD ($\bar{x}$ ± s)

**Table 2 nutrients-14-04741-t002:** Characteristics and free sugar intakes of parents (*n* = 1090).

Characteristics	Parental Free Sugar Intake Level (*n*, %)			*p*-Value
	Low (<25 g)	Middle (25~50 g)	High (>50 g)	
Total	861 (79.0)	147 (13.5)	82 (7.5)	
Gender				
Male	267 (81.9)	35 (10.7)	24 (7.4)	0.152
Female	594 (77.7)	112 (14.7)	58 (7.6)	
Ages				
≤40 year	372 (74.4)	82 (16.4)	46 (9.2)	0.003 **
41–50 year	449 (82.7)	62 (11.4)	32 (5.9)	
≥51 year	40 (85.1)	3 (6.4)	4 (8.5)	
Ethnicity				
Hans	827 (78.7)	143 (23.6)	81 (7.7)	0.182
Minorities	34 (87.2)	4 (10.3)	1 (2.6)	
Parental education				
≤Primary school	17 (68.0)	6 (24.0)	2 (8.0)	0.094
Middle school	199 (76.2)	35 (13.4)	27 (10.3)	
(Vocational) high school	365 (78.5)	62 (13.3)	38 (8.2)	
≥College	280 (82.6)	44 (13.0)	15 (4.4)	
Family monthly income				
CNY ≤ 1000	18 (90.0)	0 (0.0)	2 (10.0)	0.668
CNY 1001–3000	148 (76.7)	28 (14.5)	17 (8.8)	
CNY 3001–5000	234 (78.3)	50 (16.7)	15 (5.0)	
CNY 5001–7000	175 (78.5)	26 (11.7)	22 (9.9)	
CNY ≥7001	286 (80.6)	43 (12.1)	26 (7.3)	
BMI				
Wasting	49 (79.0)	9 (14.5)	4 (6.5)	0.364
Normal	584 (79.5)	95 (12.9)	56 (7.6)	
Overweight	198 (79.5)	34 (13.7)	17 (6.8)	
Obese	30 (68.2)	9 (20.5)	5 (11.4)	
Free sugar knowledge level (K)				
Poor	648 (78.3)	116 (14.0)	64 (7.7)	0.304
Better	213 (81.3)	31 (11.8)	18 (6.9)	
Parental attitude (A)				
Poor	424 (76.1)	84 (15.1)	49 (8.8)	0.016 *
Better	437 (82.0)	63 (11.8)	33 (6.2)	
Parental practice (P)				
Poor	611 (73.5)	141 (17.0)	79 (9.5)	<0.001 ***
Better	250 (96.5)	6 (2.3)	3 (1.2)	
Intake behavior				
Poor	263 (31.1)	217 (25.7)	366 (43.3)	<0.001 ***
Better	88 (36.1)	69 (28.3)	87 (35.7)	
Purchase and storage behavior				
Poor	278 (33.5)	219 (26.4)	334 (40.2)	0.807
Better	73 (28.2)	67 (25.9)	119 (45.9)	
Guidance behavior				
Poor	270 (30.8)	225 (25.6)	383 (43.6)	0.002 **
Better	81 (38.2)	61 (28.8)	70 (33.0)	

* *p* < 0.05, ** *p* < 0.01, *** *p* < 0.001.

**Table 3 nutrients-14-04741-t003:** Association between parental and adolescents’ free sugar intake (*n* = 1090).

Parental Free Sugar Intake	Adolescents’ Free Sugar Intake (*n*, %)			Total	r_s_	*p*-Value
	Low (<25 g)	Middle (25~50 g)	High (>50 g)			
Low (<25 g)	307 (35.7)	226 (26.2)	328 (38.1)	861	0.159	*p* < 0.001
Middle (25~50 g)	29 (19.7)	43 (29.3)	75 (51.0)	147		
High (>50 g)	15 (18.3)	17 (20.7)	50 (61.0)	82		
Total	351 (32.2)	286 (26.2)	453 (41.6)	1090		

**Table 4 nutrients-14-04741-t004:** Correlation between parental free sugar KAP and adolescents’ free sugar intake (*n* = 1090).

Parental Free Sugar KAP	Adolescents’ Free Sugar Intake (*n*, %)			r_s_	*p*-Value
	Low (<25 g)	Middle (25~50 g)	High (>50 g)		
Free sugar knowledge level (K)					
Poor	251 (30.3)	214 (25.8)	363 (43.8)	−0.087	0.004 **
Better	100 (38.2)	72 (27.5)	90 (34.4)		
Parental attitude (A)					
Poor	174 (31.2)	153 (27.5)	230 (41.3)	−0.007	0.812
Better	177 (33.2)	133 (25.0)	223 (41.8)		
Parental practice (P)					
Poor	251 (30.2)	212 (25.5)	368 (44.3)	−0.100	0.001 **
Better	100 (38.6)	74 (28.6)	85 (32.8)		
Intake behavior					
Poor	263 (31.1)	217 (25.7)	366 (43.3)	−0.062	0.040 *
Better	88 (36.1)	69 (28.3)	87 (35.7)		
Purchase and storage behavior					
Poor	278 (33.5)	219 (26.4)	334 (40.2)	0.055	0.070
Better	73 (28.2)	67 (25.9)	119 (45.9)		
Guidance behavior					
Poor	270 (30.8)	225 (25.6)	383 (43.6)	−0.085	0.005 **
Better	81 (38.2)	61 (28.8)	70 (33.0)		

* *p* < 0.05, ** *p* < 0.01.

**Table 5 nutrients-14-04741-t005:** Ordinal logistic regression model of parental and adolescents’ free sugar intake (*n* = 1090).

Variable	Model 1 ^a^ (OR, 95% CI)	Model 2 ^b^ (OR, 95% CI)	Model 3 ^c^ (OR, 95% CI)
Parental free sugar KAP (Poor = ref)			
Better knowledge level	0.720 (0.555, 0.934) *	0.717 (0.551, 0.934) ^*^	0.726 (0.557, 0.946) *
Better attitude	1.065 (0.851, 1.333)	1.058 (0.842, 1.329)	1.080 (0.859, 1.359)
Better practice	0.710 (0.467, 1.080)	0.710 (0.465, 1.085)	0.718 (0.469, 1.101)
Better intake behavior	1.203 (0.795, 1.822)	1.203 (0.790, 1.832)	1.138 (0.744, 1.740)
Better purchase and storage behavior	1.246 (0.956, 1.625)	1.257 (0.960, 1.647)	1.238 (0.944, 1.623)
Better guidance behavior	0.829 (0.613, 1.121)	0.837 (0.617, 1.134)	0.861 (0.634, 1.168)
Parental free sugar intake (Low = ref)			
Middle (25~50 g)	1.760 (1.256, 2.465) **	1.697 (1.208, 2.386) **	1.706 (1.212, 2.401) **
High (>50 g)	2.369 (1.500, 3.744) **	2.371 (1.493, 3.763) **	2.372 (1.492, 3.773) **

^a^: Model 1: parental free sugar KAP, free sugar intake. ^b^: Model 2: parental free sugar KAP, free sugar intake; parents’ gender, role, age, education level, monthly household income. ^c^: Model 3: parental free sugar KAP, free sugar intake; parents’ gender, role, age, education level, monthly household income; adjustments were made according to students’ age, grade level, and reading style. * *p* < 0.05, ** *p* < 0.01.

## Data Availability

The data that support the findings of this study are not publicly available due to the data containing information that could compromise participant privacy but are available from the corresponding author on reasonable request.

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
