# Peer review of "Is Adolescents’ Free Sugar Intake Associated with the Free Sugar Intake of Their Parents?"

_nutrients, 2022, doi:10.3390/nu14224741_

Round 1
Reviewer 1 Report
Is Adolescents' Free Sugar Intake Associated with Exposure to 2 the Free Sugar Intake of Their Parents?
A very interesting work of Zhang and colleagues, focusing on free sugar of parents an how that affects the free sugar intake of their offspring in adolescent age. Although a very nice approach, the following points have to be mentioned:
1) In line 55, it should be pointed out that the sweet preference in children and adolescents is innate and not -only- due to environmental aspects or family.
2) In chapter 3.3, the authors should rather highlight that their results -although weak positive- showed not a real association of free sugar intake of parents and adolescents. From an epidemiologic point of view, it is rather interesting that the majority of the parents and adolescents consume less free sugars than recommended by WHO (< 10 E%). That should be highlighted, at least in discussion.
3) At the beginning of discussion, the authors should not highlight that free sugar intake is high in some age groups but rather that intake is low in parents. The authors could use a recent publication by Jeanette Walton for comparison with other countries (https://pubmed.ncbi.nlm.nih.gov/34369326/).
4) Line 228-230, please include that sugar as such are not responsible for non-communicable diseases but only excess intake of calories, which is the case for all macronutrients, If you do not want to include single studies, I recommend the following review to address this issue: https://pubmed.ncbi.nlm.nih.gov/30787473/
5) Line 273- 275, there is not data available, which shows that a sugar tax prevents any kind of diseases but only affects sale or consumption, e.g., in South-Africa BMI increased after SSB-Tax introduced (https://www.sciencedirect.com/science/article/pii/S2542519620303041). So please include this article for discussion or delete the assumption that SSB-tax positively affect health.
Author Response
Dear professor,
Many thanks for your precious comments on our paper, we have revised our paper according to your comments. Please see the attachment for details of the revisions.
We would thank you again for giving us so many precious suggestions to make our study more comprehensive and more rigorous, and make our results more convincing.
Kind regards,
Lina Yang
E-mail: [email protected]

Reviewer 2 Report
In this study, the authors aimed to investigate the association between adolescents' free sugar Intake and parental free sugar intake and associated free sugar KAP. The manuscript is well-written and well-organized. However, I do have a number of comments.
1. Title, “Is Adolescents' Free Sugar Intake Associated with Exposure to the Free Sugar Intake of Their Parents?” I am not sure why you used the term “Exposure” in the title. You have determined the free sugar intake of both parents and adolescents in this study. So, it is not “exposure” to free sugar intake!
2. Please change “sugary beverages” (SSBs) to “sugar-sweetened beverages (SSBs)”.
3. Page 1, line 42, “Free sugar intake among European adolescents was higher than 87 g/d.” Please add the reference for this sentence.
4. Page 2, line 66, “Recently, a cross-sectional study was used to investigate maternal KAP of free sugar and its association with free sugar intake in children aged 6-12 years in Saudi Arabia.” Please add the reference for this sentence.
5. Page 2, line 69, “the aim of this study was to investigate the effect of parental free sugar intake and associated free sugar KAP on free sugar intake in Chinese adolescents.” This study is not investigating the effect of parental free sugar intake on adolescents' free sugar Intake, but rather it is investigating the association between parental free sugar intake and adolescents' free sugar Intake.
6. Please add the URL for reference #9 to the reference list.
7. Page 2, line 83, did you receive oral or written consent from students as well?
8. Page 2, line 84, “The two sections of the study included parent and student questionnaires, which were completed by students and their parents, respectively.” The order of the sentence is not correct. Did students complete their parents’ questionnaires?
9. Page 2, line 88, why did you exclude the schools with less than 500 students?
10. Page 3, line 107, is this semi-quantitative FFQ validated for free sugar intake?
11. Page 3, line 116, what do you mean by parents’ intake behavior? What do you mean by parents’ guidance behavior? How did you determine parents’ knowledge, attitude, and practice? Please elaborate more on this part.
12. Table 2, how did you determine parental practice and label it as poor or better?!
13. Page 6, lines 164-166, “Parents' free sugar intake was mainly concentrated in the low level (<25g/d) stage, and adolescents' free sugar intake was more in the high level (>50g/d), so the correlation analysis showed a weak positive correlation.” It is not a good justification for the observed weak association.
14. Page 5, line 162, “Parental free sugar intake levels showed a positive correlation with adolescent free sugar intake levels and the correlation coefficient was weak (rs=0.159, p<0.001).” and Page 6, line 167, “The correlation analysis of free sugar intake between parental and adolescents' free 167 sugar intake showed a positive correlation (rs=0.143, p<0.001)”. What is the difference between these two correlations?!
15. Page 6, line 174, please change “correlations” to “correlated”.
16. Page 7, line 188, please cite your previous article here.
17. Page 8, lines 223-225, “These results indicate that the sugar intake of adolescents in China shows a rising trend, and the current situation of high intake of adolescents in free sugars and added sugars deserves further attention” Is this a finding of your study? Did you look into trends of free sugar consumption?
18. Page 8, lines 225-229, please either remove these sentences or move them to the introduction section.
19. Page 8, line 250, “intake behavior had lower the free sugar intake” please remove “the”.
20. Page 9, line 267, please change “intake,” to “intake.”
21. Page 9, line 279, please specify the name of the country.
22. Page 9, line 294, “limitation of our study is that it investigated solely the Changsha area and the representative sample is limited” Do you mean the sample size is not representative of the whole Chinese population?
Author Response
Dear professor,
Many thanks for your precious comments on our paper, we have revised our paper according to your comments. Please see the attachment for details of the revisions.
We would thank you again for giving us so many precious suggestions to make our study more comprehensive and more rigorous, and make our results more convincing. At the same time, thank you for pointing out to us some details that are clearer in our methodology and conclusion section.
Kind regards,
Lina Yang
E-mail: [email protected]
